# Mechanical Simplification of Variable-Stiffness Actuators Using Dielectric Elastomer Transducers

**David P. Allen** [1,2,*], **Edgar Bolívar** [1,2], **Sophie Farmer** [1], **Walter Voit** [1,2,3] **and Robert D. Gregg** [1,2,*]

1 Department of Bioengineering, The University of Texas at Dallas, Richardson, TX 75080, USA; edgarbolivar@utdallas.edu (E.B.); Sophie.Farmer@utdallas.edu (S.F.); walter.voit@utdallas.edu (W.V.)
2 Department of Mechanical Engineering, The University of Texas at Dallas, Richardson, TX 75080, USA
3 Department of Materials Science & Engineering, The University of Texas at Dallas, Richardson, TX 75080, USA
* Correspondence: david.allen@utdallas.edu (D.P.A.); rgregg@ieee.org or rgregg@utdallas.edu (R.D.G.)

**Abstract:** Legged and gait-assistance robots can walk more efficiently if their actuators are compliant. The adjustable compliance of variable-stiffness actuators (VSAs) can enhance this benefit. However, this functionality requires additional mechanical components making VSAs impractical for some uses due to increased weight, volume, and cost. VSAs would be more practical if they could modulate the stiffness of their springs without additional components, which usually include moving parts and an additional motor. Therefore, we designed a VSA that uses dielectric elastomer transducers (DETs) for springs. It does not need mechanical stiffness-adjusting components because DETs soften due to electrostatic forces. This paper presents details and performance of our design. Our DET VSA demonstrated independent modulation of its equilibrium position and stiffness. Our design approach could make it practical to obtain the benefits of variable-stiffness actuation with less weight, volume, and cost than normally accompanies them, once weaknesses of DET technology are addressed.

**Keywords:** variable-stiffness actuator; dielectric elastomer transducer; dielectric elastomer actuator

## 1. Introduction

Compliant actuation can benefit many robots, especially if the compliance is adjustable. In particular, legged and gait-assistance robots can walk more efficiently with compliant actuators [1–3]. Adjusting their actuator compliance [4] could help them adapt to variations in gait speed and type [5]. However, the additional mechanical components that a variable-stiffness mechanism adds to an actuator makes variable-stiffness actuators (VSAs) impractical for some uses [2]. These components increase the VSA's weight, volume, and cost compared to rigid actuators and fixed-stiffness series elastic actuators [2]. In this work, we show how to accomplish VSA functionality without the mechanical complexity that VSAs normally entail by using dielectric elastomer transducers (DETs) as the core of a VSA's variable-stiffness mechanism.

Two recent VSAs are examples of the mechanical complexity of state-of-the-art VSA design. First, the ARES-XL [6] is designed for use in a gait-assistance exoskeleton for rehabilitation. Its variable-stiffness mechanism has four major component motions when it is deflected: two rotations and two sliding contacts. Second, a recent version of the MACCEPA has been used in a gait rehabilitation exoskeleton [7]. This actuator also has four major component motions in its variable-stiffness mechanism: two rotations, a chain bending, and a sliding contact. The stiffness-adjusting motors and other moving components of these two VSAs add considerably to their size and weight. Though other

VSAs use a variety of variable-stiffness mechanisms [8,9], similar observations could be made about their mechanical complexity.

Low power stiffness modulation was achieved without the complexity of VSAs by a positive-negative-stiffness actuator [10] and an electroadhesive clutch and spring mechanism [11], but they did not have all the functionality of a VSA. The positive-negative-stiffness actuator needed only one motor to control its position and inherent stiffness, making it simpler than state-of-the-art VSAs. In experiments, it consumed 3 W while modulating its stiffness under load. However, this actuator relied on the bifurcation of its mechanism's behavior to switch between stiffness and position control, so it could not control its equilibrium position and stiffness independently. The electroadhesive clutch and spring mechanism [11] increased its stiffness by a factor of 36 and consumed an average of 0.6 mW during operation in an ankle exoskeleton. However, the nature of this device limits it to a set of discrete stiffness values rather than the continuous range of values exhibited by state-of-the-art VSAs, and it cannot modulate its equilibrium position without an external load. Both devices could be useful in robotic applications, but they cannot substitute generally for a full-featured VSA.

DETs are softening polymer devices that have been used to make compliant actuators, but not a fully functioning VSA before this work. Prior DET compliant actuators can control their equilibrium position (the output position when no load is applied) and inherent stiffness, but not independently and simultaneously. DET diaphragm modules, developed for variable-stiffness suspensions, can vary their stiffness but not their equilibrium position [12–14]. Coupling one or more diaphragm modules with a biasing mechanism [15] results in a dielectric elastomer actuator that has one degree of freedom. Such an actuator changes both its stiffness and equilibrium position, but these two changes are coupled. Using a second DET diaphragm module as the biasing mechanism [16,17] adds a second degree of freedom, but only partially decouples the control of stiffness and equilibrium position. A DET orthosis [18] can vary its stiffness and equilibrium independently, but it does so by using closed-loop control rather than modulating its inherent stiffness. A prior DET VSA design by some of the authors [19] sought independent control of stiffness and equilibrium position, but it never achieved stiffness modulation because its DETs were impractical to manufacture reliably.

Our new DET VSA can control its equilibrium position and inherent stiffness independently and simultaneously without the mechanical complexity of state-of-the-art VSAs. As explained in Section 2, our use of cone-diaphragm DETs in a VSA addresses two design challenges: (1) the shift of VSA equilibrium position when the DETs change stiffness, and (2) the need to maintain tension in the DETs' elastomer films. Our design approach also presents a means to use DETs with greater forces and displacements than they can generate themselves. The mechanical model reviewed in Section 3 is used to derive analytical formulas that can be used to determine how the dimensions of a DET module affect its uncharged stiffness, voltage-induced stiffness change, and maximum displacement. The electrical model also reviewed in that section explains the sources of electrical energy losses in DETs. Experimental results discussed in Section 4 confirm the feasibility of our design approach, which uses DETs with larger forces than typically present in prior works. Specifically, the results cover our actuator's stiffness change magnitude and speed, viscoelasticity at varying speeds, and electrical power requirements for stiffness change. The experimental methods used to investigate our DET VSA's functionality are given in Appendix A.

## 2. Design

The design of our DET VSA achieves variable-stiffness actuation without the mechanical complexity of state-of-the-art VSAs because it does not need auxiliary mechanical components to modulate the inherent stiffness of its elastic components. As this section explains, such stiffness modulation is possible because DETs soften due to electrostatic forces. To implement DET stiffness modulation, the design must account for the equilibrium point shift of DETs and the inability of elastomer films to support tension. It does so by using cone-diaphragm DET modules, which do not change equilibrium point, and which keep their elastomer films tensioned regardless of the load

direction. The design achieves independent control of stiffness and equilibrium position because it uses an electric motor to control the VSA's equilibrium position and DETs to control the VSA's stiffness. This arrangement results in a variable-stiffness mechanism that has only one component motion, no rolling or sliding components, and no stiffness-adjusting motor.

DETs soften due to electrostatic forces. Essentially, a DET is a thin film of dielectric elastomer coated on its top and bottom faces with stretchable electrodes [20,21] (Figure 1). A constant voltage $\Phi$ applied across the electrodes decreases the stiffness of the DET in proportion to the square of the voltage [21,22]. This softening occurs because the voltage causes the DET to store charges such as a parallel-plate capacitor, and these charges exert electrostatic forces on the elastomer film that tend to expand it in area and compress it in thickness. As the DET expands and thins, its capacitance increases. If it is also subject to a constant voltage during expansion, more charge flows onto its electrodes strengthening the electrostatic forces. Thus, the constant-voltage electrostatic forces act as a negative-stiffness mechanism that counteracts the elastomer's elastic restoring forces, making the DET softer. When no voltage is applied, the DET defaults to a (relatively) stiff state.

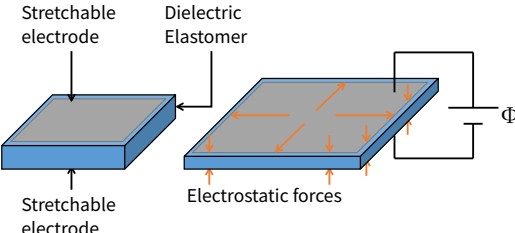

**Figure 1.** Working principle of a dielectric elastomer transducer (DET). A DET, consisting of a thin film of dielectric elastomer coated with stretchable electrodes, softens when a constant voltage $\Phi$ is applied across its electrodes due to electrostatic forces that tend to expand it in area and compress it in thickness.

The nature of DETs poses two challenges for the design of a DET variable-stiffness mechanism. First, the electrostatic forces that soften DETs cause some types of DETs to expand when charged. This expansion could cause the equilibrium position of a variable-stiffness mechanism to change when its stiffness changes, which is typically undesirable. Second, an elastomer film typically cannot support compression in its planar directions. This characteristic complicates the design of variable-stiffness mechanisms that must support compression and tension.

The cone-diaphragm DET configuration used in our VSA design (and some previous variable-stiffness devices [12–14]) addresses these challenges. In this configuration, a pre-stretched, adhesive elastomer (VHB 4910) connects center disks to an outer frame (Figure 2). The elastomer is coated on its top and bottom faces with conductive graphite powder, which forms the electrodes of the DET. Polyimide film reinforces the elastomer against the electric field and mechanical stress concentrations that occur at the edges of the electrodes. During operation, the center disks displace out of plane, like the motion of the center of a speaker cone, stretching the elastomer. The out-of-plane motion decouples the DET's equilibrium position from its stiffness. Additionally, it enables the DET to support bidirectional loads because a displacement in either direction tensions the elastomer film. Pairing other DET configurations together in an antagonistic pair as in our previous work [19] and other works [23] is another solution to these challenges. However, this solution requires additional design work to ensure that the antagonistic DETs remain in tension throughout the mechanism's range of motion.

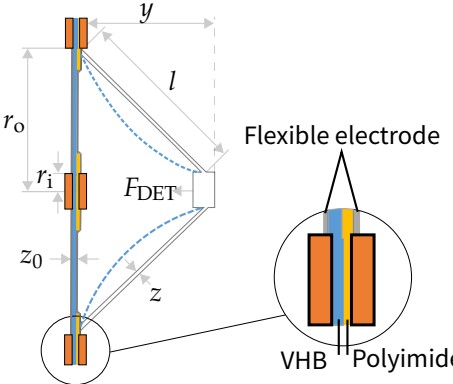
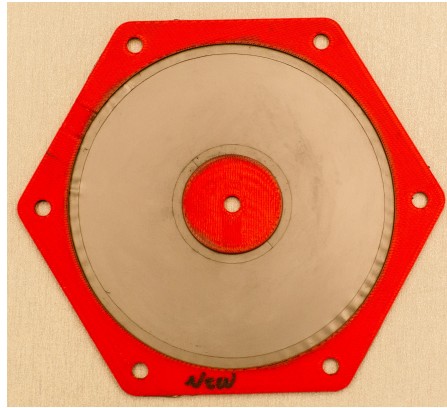

**Figure 2.** Cone-diaphragm DET configuration. The DETs used in this work were cone-diaphragm modules as depicted in photograph on the right. When a module's center disk is displaced, the elastomer film deforms into the curved-cone shape depicted by the dashed lines in the left diagram. For modeling purposes, the deformed shape is approximated as a straight-sided cone depicted by the solid outline in the left diagram.

Our DET VSA (Figure 3) achieves independent and simultaneous control of stiffness and equilibrium position, like other VSAs, and is a linear actuator to fit the linear motion of DETs. A direct drive ball screw converts the rotation of the motor (EC45 flat, 70 W, Maxon Precision Motors, Taunton, MA, USA) into linear motion and connects the motor to the variable-stiffness mechanism. In this arrangement, the motor sets the equilibrium position of the VSA and supplies the force to maintain that position. The variable-stiffness mechanism controls the actuator's stiffness and transmits the force to the load. The variable-stiffness mechanism consists of a stack of 30 cone-diaphragm DET modules, capped on the ends by two insulating cone-diaphragm modules. The modules' center disks are connected to the VSA's ball screw, and the module frames are connected to the actuator's output as shown in Figure 3. Thus, the DET modules add their force together when stretched, so the force and stiffness of the VSA are linearly proportional to the number of modules installed in the variable-stiffness mechanism. Electrically, the DET modules are connected in parallel, so they all charge and discharge together. A linear-bearing guide-rod system serves as the backbone of the actuator maintaining its components in alignment. The dimensions of our DET VSA are given in Table 1.

**Table 1.** Dimensions of our DET VSA and its DET modules.

| VSA | |
| --- | --- |
| Max. length | 450 mm |
| Output position travel | 90 mm |
| Equilibrium position travel | 42 mm |
| Width | 134 mm |
| Height | 108 mm |
| Mass | 880 g |
| **DET Module** | |
| $r_o$ | 41.3 mm |
| $r_i$ | 12.7 mm |
| $y_{Max}$ | See Section 4.6 |
| $z_0$ | 63 μm |

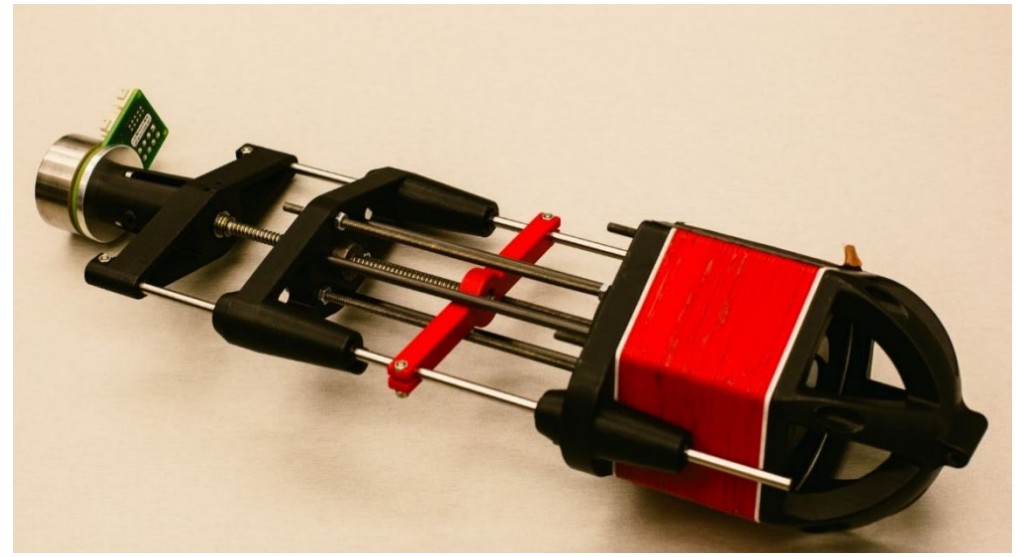

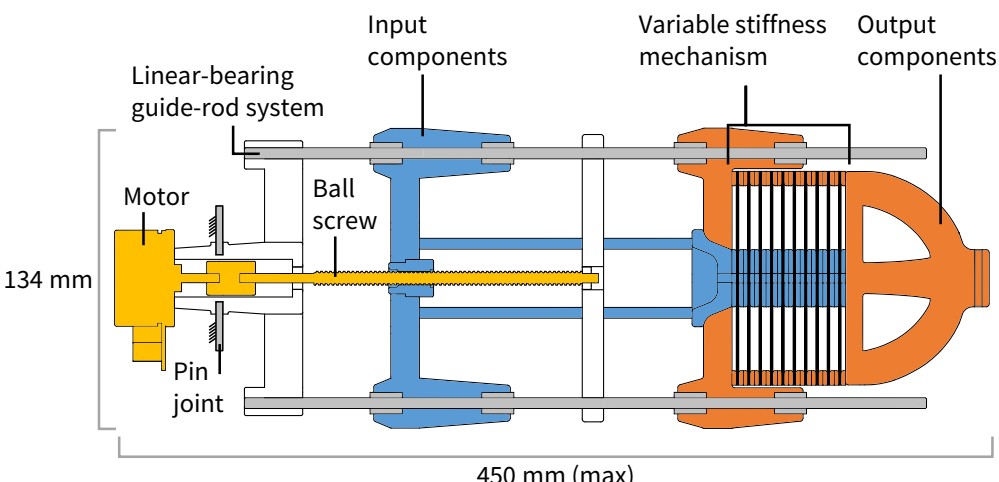

**Figure 3.** Our DET variable-stiffness actuator (VSA). This actuator has only one component motion in its variable-stiffness mechanism: the relative translation of the DET modules' center disks and outer frames. The modules' center disks are connected to the input components (blue), which are driven by the VSA's motor and ball screw (yellow). The modules' outer frames are connected to the actuator's load by the output components (orange). Both the input and output components are constrained to linear motion by the linear-bearing guide-rod system. The variable-stiffness mechanism consists of 32 DET modules (30 active, and two insulating) though the diagram shows only ten for clarity. The variable-stiffness mechanism softens when the DET modules are charged with constant voltage.

The use of DETs makes the variable-stiffness mechanism mechanically simpler than those of other VSAs. The DET variable-stiffness mechanism has merely one major component motion: linear displacement of the center disks that stretches the DET modules. In contrast, the VSA variable-stiffness mechanisms mentioned in Section 1 [6,7] have four component motions. Rather than the variety of components in prior variable-stiffness mechanisms, our design has only one type of component: the DET modules, the number of which can be selected to fit the force and stiffness needs of the application. Because the variable-stiffness mechanism has no rolling or sliding components, it does not need any bearings or bushings, simplifying maintenance. Finally, the VSA does not need an additional motor to control its stiffness, as many others do, because its stiffness is controlled by a voltage input.

The hybrid combination of an electric motor and a DET variable-stiffness mechanism simplifies our VSA's control scheme and benefits from the strengths of each transducer. Because this approach restricts each transducer to one task: the motor to equilibrium position modulation and the DETs

to stiffness modulation, it fundamentally decouples the two functions simplifying the VSA's control scheme. In this arrangement, the two transducers complement each other. The motor readily generates large forces and motions and can be operated with simple position control methods. The DETs provide elasticity and stiffness modulation with a simple voltage input, while operating with higher forces and strokes than they could produce as prime movers themselves.

Because the DET modules soften when charged with a constant voltage, the DET VSA is stiffest by default, which could be advantageous for robotic prostheses and orthoses. Robotic prostheses and orthoses for legs should default to stiff settings when they lose power to maintain support for their wearer. They may also need to maintain stiff settings for long periods when their wearer is standing still, which could be energetically costly for a VSA that requires energy to remain stiff. VSAs that do not require power to maintain stiffness [24,25] could also be efficient in this application.

## 3. Modeling of Variable-Stiffness Module

This section first reviews a DET electrical model and a cone-diaphragm mechanical model from other works [26–28]. It uses the electrical model to give insight into the causes of electrical energy storage and losses in DETs that will be used to interpret the results in Section 4. It then uses the mechanical model to derive the effects of module dimensions on a module's uncharged stiffness, voltage-induced stiffness change, and maximum displacement. Knowledge of these effects will be useful for adapting our DET VSA's performance to specific applications.

### 3.1. Electrical

An electrical circuit consisting of a capacitance $C$ with a series resistance $R_s$ and parallel resistance $R_p$ [26,27] (Figure 4) is a model for the electrical energy stored and dissipated in a DET. Because a DET consists of a pair of electrodes separated by a dielectric, it capacitively stores electrical energy $U_c$ according to

$$U_c = \frac{1}{2}C\Phi_{DET}^2, \tag{1}$$

where $\Phi_{DET}$ is the voltage on the DET capacitance. The series resistance represents the electrical resistance of the DET's electrodes, which dissipates energy through Joule heating. The energy dissipated during charging or discharging is proportional to the charge rate and series resistance value [29]. The parallel resistance represents the pathway for leakage current through the dielectric membrane. This current dissipates power $P$ through Joule heating in the dielectric according to

$$P = \frac{\Phi_{DET}^2}{R_p}. \tag{2}$$

As the DET is displaced, its electrode area and membrane thickness change causing its capacitance and parallel resistance to change. The series resistance also changes as the DET deforms, but this change is typically negligible being much smaller than the changes in capacitance and parallel resistance [26].

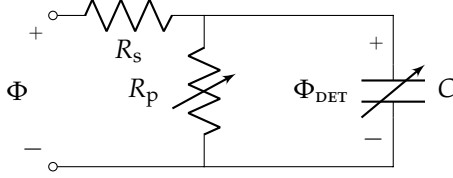

**Figure 4.** Electrical model of a DET. The electrical behavior of a DET is modeled in this work by a capacitance $C$ that represents the charge storage of the DET that changes as the DET deforms, with a series resistance $R_s$ that represents the resistance in the DET's electrodes, and a parallel resistance $R_p$ that represents the current leakage path through the dielectric elastomer. Schematic redrawn and modified from [26].

### 3.2. Mechanical

Displacement of the variable-stiffness mechanism deforms the membrane of an individual DET module into a curved-cone shape that is reasonably approximated as a straight-sided truncated cone (Figure 2) [12,28]. In this model, when the variable-stiffness mechanism is displaced by $y$, the membrane takes the shape of a truncated cone with slant height $l$ and thickness $z$. When undeformed, the membrane is shaped like an annulus with outer and inner radii $r_\mathrm{o}$ and $r_\mathrm{i}$ and thickness $z_0$, and the slant height $l$ reduces to $l_0 = r_\mathrm{o} - r_\mathrm{i}$. The radial-direction stretch $\lambda_\mathrm{r}$, obtained using the Pythagorean theorem, is

$$\lambda_\mathrm{r} = \lambda_\mathrm{P} \frac{l}{l_0} = \lambda_\mathrm{P} \sqrt{1 + \left(\frac{y}{l_0}\right)^2}, \tag{3}$$

where $\lambda_\mathrm{p}$ is the prestretch of the membrane [26,28]. The truncated-cone model implies that the circumferential-direction stretch $\lambda_\mathrm{c}$ is equal to the constant prestretch $\lambda_\mathrm{p}$ and that the membrane's radial stretch and thickness are homogeneous. In reality, the membrane's stretches and thickness are inhomogeneous as shown by more accurate analyses using numerical solutions of coupled nonlinear differential and algebraic equations [30] and finite element analysis [13,14]. However, we chose to use the truncated-cone model because it simplifies the derivation of the effects of module dimensions given in Section 3.3 and predicts module behavior with a useful level of accuracy [28] without the computational burden of the more accurate methods.

The net force exerted by a DET module $F_\mathrm{DET}$ is caused by the radial-direction stress $\sigma_\mathrm{r}$ in the module's membrane. With the assumption that the membrane has constant volume, the relation between DET force and material stress is

$$F_\mathrm{DET} = \sigma_\mathrm{r} \pi (l_0 + 2r_\mathrm{i}) l_0 z_0 \frac{y}{l_0^2 + y^2} \tag{4}$$

as seen from references [26,28]. The material stress is the sum of hyperelastic, electrical, and viscoelastic stresses. For a DET with constant volume, constant permittivity, and a single mechanical degree of freedom oriented perpendicular to the electric field, material stress in the actuation direction is given by

$$\sigma_\mathrm{r} = \underbrace{\lambda_\mathrm{r} \frac{\partial \psi_\mathrm{h}}{\partial \lambda_\mathrm{r}}}_{\text{hyperelastic}} \underbrace{- \epsilon_0 \epsilon_\mathrm{r} E^2}_{\text{electrical}} + \underbrace{\sum_{i=1}^{n} k_i (\lambda_\mathrm{r} - 1 - \xi_i) + \eta_{n+1} \dot{\lambda}_\mathrm{r}}_{\text{viscoelastic}}, \tag{5}$$

where $\psi_\mathrm{h}$ is a hyperelastic energy density function, $\epsilon_0$ is the permittivity of free space, $\epsilon_\mathrm{r}$ is the dielectric elastomer's relative permittivity, $E$ is the electric field applied across the dielectric elastomer, $\eta_i$ are damping coefficients, $k_i$ are spring stiffnesses, and $\xi_i$ are damper strains, with $\xi_{n+1} = \lambda_\mathrm{r} - 1$ [26]. Though a cone diaphragm displaces out of plane, this model is applicable because the cone diaphragm's material stretch is perpendicular to the electric field applied across its membrane.

### 3.3. Effect of Module Dimensions

This subsection discusses the effect of module dimensions ($r_\mathrm{o}$ and $r_\mathrm{i}$) on a cone-diaphragm module's uncharged stiffness, voltage-induced stiffness change, and maximum displacement.

This work defines the stiffness of a DET module to be $F_\mathrm{DET}/y$ rather than the standard definition $\partial F_\mathrm{DET}/\partial y$ because this definition captures the change in force at a given displacement caused by the adjustment of a variable-stiffness mechanism. Accordingly, in this work, stiffness means

$$\frac{\Delta F_\mathrm{DET}}{\Delta y} = \frac{F_\mathrm{DET}(y) - F_\mathrm{DET}(y = 0)}{y - 0} = \frac{F_\mathrm{DET}}{y}. \tag{6}$$

This metric can be understood as a linear approximation of the net effect of nonlinear behavior over a finite region.

The effect of module dimensions on a cone-diaphragm's stiffness can be calculated by combining Equations (4) and (5) to get

$$\frac{F_{\text{DET}}}{y} = \frac{1}{y}\left[\left(\lambda_{\text{r}}\frac{\partial \psi_{\text{h}}}{\partial \lambda_{\text{r}}} - \epsilon_0\epsilon_{\text{r}}E^2\right)\pi(l_0 + 2r_{\text{i}})l_0 z_0 \frac{y}{l_0^2 + y^2}\right],\tag{7}$$

where the effects of viscoelasticity have been neglected. Proceeding further requires a choice for the strain-energy density function $\psi_{\text{h}}$. Let $\psi_{\text{h}}$ be the Neo-Hookean strain-energy density function for simplicity. Simplified for a constant volume cone-diaphragm DET, this function is

$$\psi_{\text{h}} = \frac{\mu}{2}\left(\lambda_{\text{r}}^2 + \lambda_{\text{p}}^2 + (\lambda_{\text{r}}\lambda_{\text{p}})^{-2} - 3\right).\tag{8}$$

Substituting Equation (8) into Equation (7) and simplifying yields the DET module's stiffness:

$$\frac{F_{\text{DET}}}{y} = \frac{\pi(l_0 + 2r_{\text{i}})z_0}{l_0}\left[\mu\left(\lambda_{\text{p}}^2 + \frac{l_0^4}{\lambda_{\text{p}}^4\left(l_0^2 + y^2\right)^2}\right) - \frac{\epsilon_0\epsilon_{\text{r}}\Phi_{\text{DET}}^2}{z_0^2}\right].\tag{9}$$

A similar expression can be derived using the standard definition of stiffness $\partial F_{\text{DET}}/\partial y$. Accordingly, the module's uncharged stiffness (when $\Phi_{\text{DET}} = 0$) varies with its dimensions as shown in Table 2.

**Table 2.** How module dimensions affect the performance of cone-diaphragm modules.

|  | Constant | Increasing | Uncharged Stiffness | Voltage-Induced Stiffness Change | Maximum Displacement |
|---|---|---|---|---|---|
| Case 1 | $l_0$ | $r_{\text{i}}$ | increases | increases | is constant |
| Case 2 | $r_{\text{i}}$ | $r_{\text{o}}$ | undetermined | decreases | increases |
| Case 3 | $r_{\text{o}}$ | $r_{\text{i}}$ | undetermined | increases | decreases |

The effect of module dimensions on a cone-diaphragm's voltage-induced stiffness change can be determined from Equation (9) as

$$\frac{\partial}{\partial \Phi}\frac{F_{\text{DET}}}{y} = -\frac{2\epsilon_0\epsilon_{\text{r}}\pi\Phi}{z_0}\left(\frac{2r_{\text{i}}}{l_0} + 1\right),\tag{10}$$

where electrical dynamics are neglected by assuming $\Phi_{\text{DET}} = \Phi$ and viscoelasticity is neglected by setting $k_i = 0$ and $\eta_{n+1} = 0$. According to this equation, the voltage-induced stiffness change varies with $r_{\text{o}}$ and $r_{\text{i}}$ as reported in Table 2.

The effect of module dimensions on a cone diaphragm's maximum displacement can be calculated by substituting the maximum stretch for the membrane material $\lambda_{\text{r, Max}}$ into Equation (3) and solving for $y$:

$$y_{\text{Max}} = l_0\sqrt{\left(\frac{\lambda_{\text{r, Max}}}{\lambda_{\text{p}}}\right)^2 - 1}.\tag{11}$$

According to this equation, the maximum displacement of a cone-diaphragm module varies with $r_{\text{o}}$ and $r_{\text{i}}$ as reported in Table 2.

## 4. Results and Discussion

This section reports the functionality of our DET VSA: its independent and simultaneous modulation of stiffness and equilibrium position, and the magnitude and speed of its stiffness change.

Then, it reports on characteristics of the variable-stiffness mechanism relevant to determining its efficiency: its viscoelasticity and the electrical energy and power required to change its stiffness. Finally, the section discusses the maximum displacement achievable by the variable-stiffness mechanism, and potential solutions for weaknesses of DET technology. Details of the test procedures are given in Appendix A.

### 4.1. Modulation of Stiffness and Equilibrium Position

A force-displacement plot of a VSA's behavior can show the coupling between a VSA's stiffness and equilibrium position. A suitable plot is generated by fixing a VSA's equilibrium position, perturbing the VSA's output point with a range of stiffness settings, and repeating these steps for additional equilibrium positions. The stiffnesses of the VSA appear in this plot as the slopes of the curves generated with the equilibrium position fixed, and the equilibrium positions appear as displacement values where force is zero. If the VSA's stiffness and equilibrium position are coupled, then the VSA cannot display a full range of stiffnesses at all equilibrium positions.

Our DET VSA can modulate its stiffness and equilibrium position independently. A force-displacement plot generated by our DET VSA (Figure 5) has a pair of stiffer and softer curves that originate from equilibrium position 1 at 0 mm. The stiffer curve was generated with our VSA's variable-stiffness mechanism discharged, and the softer curve with the mechanism charged to 5.0 kV. These curves show that the VSA can modulate its stiffness. The VSA can also have intermediate stiffness values, as discussed in Section 4.2. Our VSA's force-displacement plot has a range of zero-force points between equilibrium positions 1 and 2 that was generated by using the actuation motor to shift the VSA's equilibrium position. This feature shows that the VSA can control its equilibrium position. Finally, the signature has another pair of stiff and soft force-displacement trajectories that originate from equilibrium position 2. These curves are identical to those originating from equilibrium position 1, so the plot shows that the VSA can reach its full range of stiffnesses across its range of equilibrium positions.

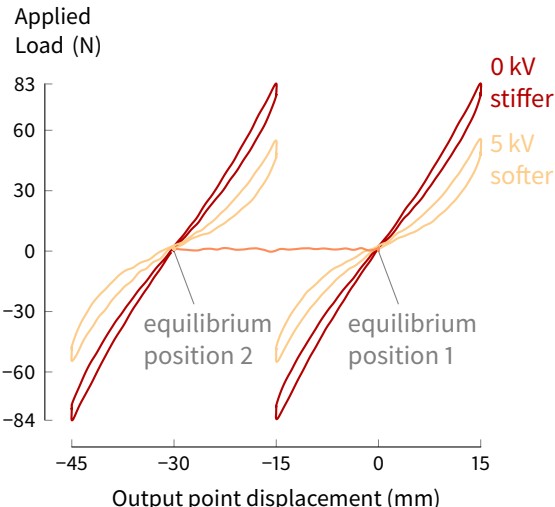

**Figure 5.** Our DET VSA can independently modulate its stiffness and equilibrium position as seen in this plot of the load applied to the DET VSA and its output point displacement. It has the same range of stiffnesses after the actuation motor shifted the equilibrium position from equilibrium position 1 to equilibrium position 2.

### 4.2. Stiffness Change Magnitude

We calculated the stiffness change from force-displacement data from tensile tests (Figure 6). In these tests, a testbed displaced the VSA's output point while the VSA's actuation motor maintained a constant equilibrium position for three compression-tension cycles. In keeping with our definition of

stiffness in Section 3.3, stiffness was calculated as $F_{DET}/y$ for [5, 15, 25] mm. The corresponding stiffness changes in Figure 7 were determined from the rising tension portion of the third cycle.

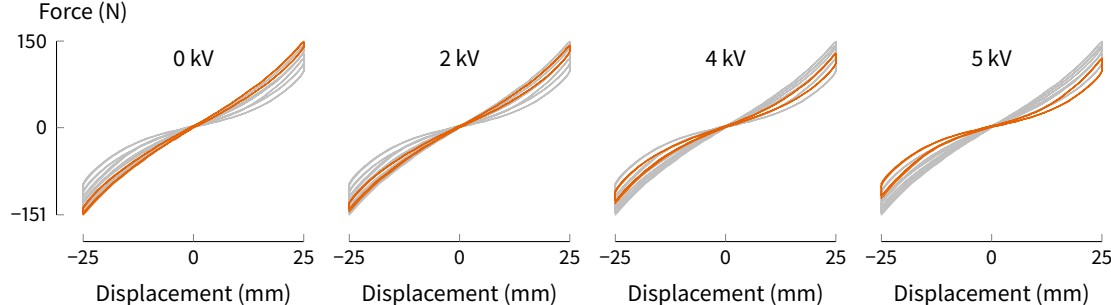

**Figure 6.** The VSA's variable stiffness mechanism softens when charged with a constant voltage, as seen in these tensile test results with a displacement rate of 1 mm/s. The loops progress in a clockwise direction. The light gray curves are data from the other voltage levels given for context.

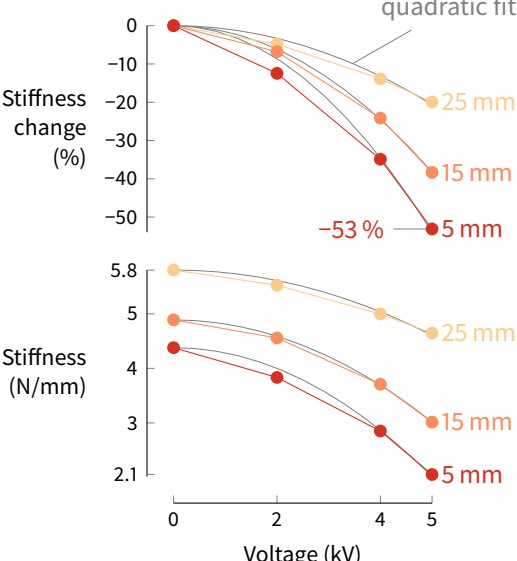

**Figure 7.** Our DET VSA softened approximately quadratically with the applied voltage up to 53 %. These values were calculated from the rising tension portion of the third cycles of the trajectories shown in Figure 6 by dividing the force at [5, 15, 25] mm by the corresponding displacement. Because the DET stiffness is quadratically dependent on voltage, quadratic fits using only a quadratic term and a constant term equal to the initial point are plotted for comparison with the experimental data.

As the voltage on the DET modules increased, our DET VSA softened approximately quadratically up to 53 % (Figure 7). Quadratic fits (using only a quadratic term and a constant term equal to the initial point) are plotted for comparison with the experimental data because the DET stiffness is quadratically dependent on voltage (Equation (9)). Though we only report data at discrete voltage levels, the VSA can have a continuous range of stiffness levels because the stiffness change is controlled by voltage input, which can have a continuous range of values.

The 53 % stiffness reduction of our DET is not a fundamental limit of the design, and greater stiffness changes are possible. Because the stiffness change increases dramatically with the DET voltage, even a slightly higher voltage can noticeably increase the stiffness change. While the elastomer in our DET modules (VHB 4910 with a 400 % biaxial prestretch) can sustain 15 kV when the energized region is small [31], under more relevant test conditions its breakdown voltage is about 6 kV [32]. In the development of our DET VSA, we tested many prototype DET modules at 6 kV, and these tests yielded greater stiffness reductions than those at 5 kV. DET diaphragm modules with different proportions

and 385 % biaxial prestretch have displayed zero stiffness (100 % reduction) for small displacements when charged with 6.25 kV [14].

### 4.3. Stiffness Change Speed

To measure how fast the VSA could change stiffness, the variable-stiffness mechanism was charged to 5 kV with displacements of [5, 15, 25] mm. The DET modules used in the variable-stiffness mechanism can charge rapidly because they are capacitors, but the high-voltage power supply used in this work could not supply enough current to charge all of them simultaneously at their maximum rate. Accordingly, when testing to see how rapidly the VSA could soften, only three DET modules were installed on the VSA. Stiffness was calculated as force divided by displacement consistent with the definition in Section 3.3. This method is also consistent with that used to measure the speed of stiffness change in reference [14].

The VSA's stiffness change can be rapid (within 50 ms), because it is limited primarily by charge rate, but it is also limited by a slow-decay mechanism. During testing, the VSA's stiffness dropped rapidly and then decayed slowly (Figure 8). The rapid drop corresponds to the period when the DETs were charging (0 ms to 40 ms), so the power supply's performance limits the stiffness change during this period. The slow decay was short for 5 mm displacement, so that the stiffness changed within 50 ms. In another work [14], a smaller cone-diaphragm module softened in 12 ms when charged with a power supply with a faster voltage change rate than that of our power supply (DCH3320P1, High-Voltage Power Solutions).

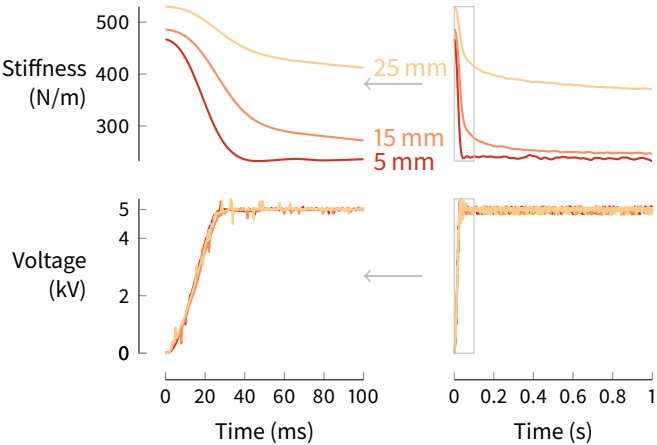

**Figure 8.** The DET VSA changed stiffness rapidly while charging to 5 kV with only three DET modules installed and the variable-stiffness mechanism's displacement fixed at [5, 15, 25] mm.

The slow decay that follows the rapid drop does not appear to be caused directly by electrostatic forces because the voltage is steady during the slow decay period. We suggest that the slow decay may be caused by viscoelastic relaxation within the membrane. The application of voltage to a module's electrodes may cause its elastomer membrane's deformed shape to become more curved and reduce the force exerted by the module. The deviation between the actual curved-cone shape of a displaced cone-diaphragm module and the straight-sided cone approximation increases with displacement $y$, so this effect should be more prominent for larger displacements. Furthermore, this voltage-induced curvature would likely be damped by the viscoelasticity of the elastomer membranes, resulting in the slow decay of stiffness seen in the stiffness-change speed tests (Figure 8).

### 4.4. Viscoelasticity

The viscoelastic behavior of the DET modules had a significant influence on the DET VSA's force and stiffness (Figure 9). Tensile tests with greater displacement rates yielded greater forces and more hysteresis. The viscoelasticity of the dielectric elastomer used in the DET modules explains these two

effects because viscosity damps motion [33]. Viscoelasticity also caused forces to be highest on the first cycle and to be lower on subsequent cycles as the DETs settled into a steady-state response. This effect is most obvious for the 100 mm/s trials, but it occurred for all measured data. The viscoelastic hysteresis was quantified by integrating the third tension cycle of the 0 kV curves in Figure 9, yielding the mechanical energy absorbed and returned by the variable-stiffness mechanism (Figure 10). The difference between the energy absorbed and returned is the energy dissipated by viscoelasticity.

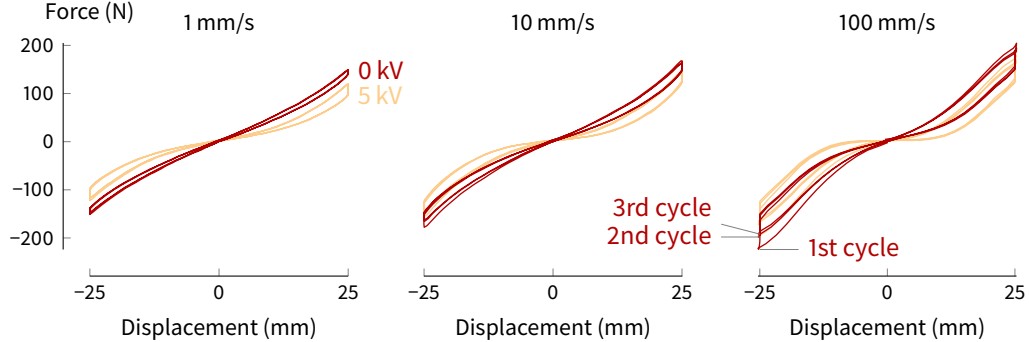

**Figure 9.** The viscoelasticity of the DET modules caused them to exert more force and have more hysteresis when the variable-stiffness mechanism was displaced more rapidly. The force-displacement trajectories settled towards a steady-state response producing peak forces of less magnitude on each subsequent cycle. The loops progress in a clockwise direction, so the hysteresis is dissipative.

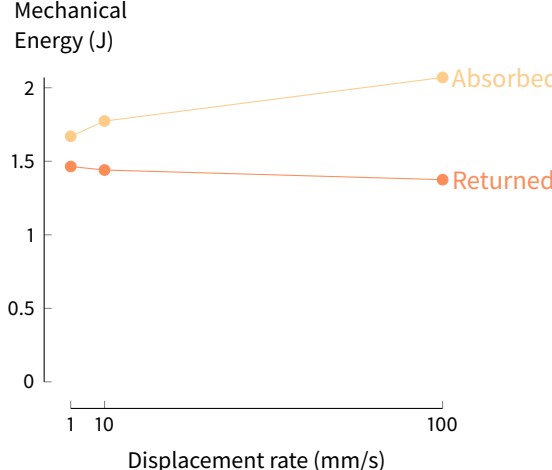

**Figure 10.** Our DET VSA absorbed more and returned less mechanical energy at faster displacement rates. Thus, it dissipated more mechanical energy at faster displacement rates. These values were calculated by integrating the third tension cycle of the 0 kV curves in Figure 9.

### 4.5. Electrical Power Requirements for Stiffness Change

Most of the electrical energy used to change stiffness is stored in the DET modules, and much of this stored energy could be recovered with appropriate circuitry. We measured the energy used to change stiffness by integrating the electrical power supplied to the variable-stiffness mechanism during relaxation tests, wherein the VSA was held at a constant displacement while its variable-stiffness mechanism was charged. The energy stored in the DET modules during the relaxation tests according to Equation (1) is only a little less than the experimentally measured energy used to change stiffness (Figure 11) and could potentially be recovered.

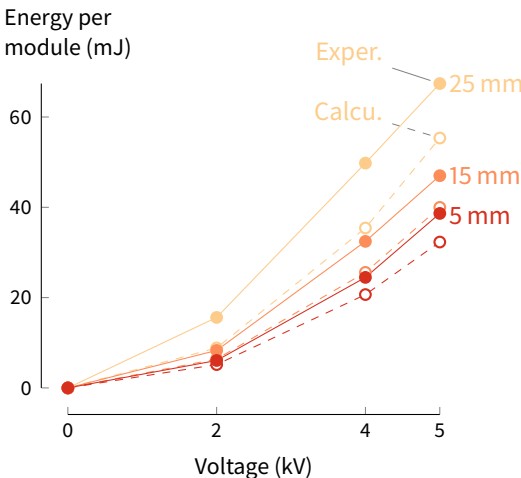

**Figure 11.** Much of the electrical energy used to charge the DET modules was stored in the modules and could be recovered to increase the efficiency of stiffness changing. These values are the energy used to change stiffness during relaxation tests. The stored energy values calculated from Equation (1) are marked with hollow markers connected by dashed lines, and experimental values are shown with circular markers connected by solid lines.

Our DET VSA requires continuous electrical power to hold a reduced stiffness due to current leakage through the dielectric. To determine the amount of power required, we measured the leakage current through five DET modules connected in parallel for 6 min and multiplied the measurement by the supply voltage (Figure 12). The current decreased with time, and the leakage power at 3 min was [0.06, 1.33, 5.15] mW for [2, 4, 5] kV. Using these power and voltage values in Equation (2) to calculate the leakage resistance yields [67, 12, 5] GΩ. The theoretical power consumption by constant resistances of these values according to Equation (2) are plotted in Figure 12 to make it clear that the leakage power of the DETs is not well represented by a constant resistance. This behavior can be explained by the decrease of VHB 4910's volume resistivity under high electric fields [34]. The change in leakage current over time can be explained by the process of dielectric absorption [35].

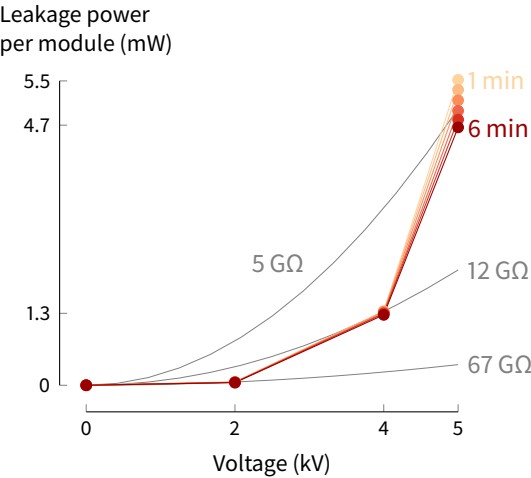

**Figure 12.** Current leakage through the dielectric causes our DET VSA to draw continuous electrical power while holding reduced stiffness as determined from measurements of the leakage current through five DET modules connected in parallel. The leakage power we measured was not consistent with the power dissipated through a constant resistance value (shown by the gray curves). The leakage power decreased over time while the DETs were held at constant voltage.

### 4.6. Maximum Displacement of Variable-Stiffness Mechanism

In the experiments in this work, displacements of the variable-stiffness mechanism were limited to 25 mm to avoid failure, but previous work and our experience indicate that greater displacement is possible. The state of stress in a cone-diaphragm DET is similar to that in a "pure shear" DET [28]. Such a DET made with VHB 4905 has been actuated to a state of 700 % × 400 % stretch [36]. In a uniaxial tensile test, a strip of VHB 4910 endured over 1200 % stretch before breaking, but in relaxation tests, strips of VHB 4910 broke while being held at 700 % stretch [37]. Therefore, it is reasonable to assume a failure stretch of 700 % for VHB 4910. Substituting $\lambda_{r,\,Max} = 700\,\%$ into Equation (11) yields a maximum displacement for the variable-stiffness mechanism $y_{Max} = 41$ mm. This calculation fits with our observations that 42.3 mm to 61.6 mm was the range of failure displacements for prototype cone-diaphragm modules with the same dimensions as those used in this work.

### 4.7. Discussion of Solutions for DET Weaknesses

Other research discussed in this section has reported potential solutions for the weaknesses of DET technology: poor reliability, viscosity, and the need for high voltages.

The biggest challenge in this work was achieving reliable operation of the DET modules. Several modules initially worked well, but later failed with less than a thousand motion cycles after being stored for about two months. Reliability can be improved through proper choice of electrode formulation, operating in a dry environment (<5% relative humidity), or operating at low electric fields [38]. Some DETs have operated for millions of cycles [38]. Cone diaphragms in particular have achieved tens of thousands of cycles before failure [14].

Some applications may benefit from the viscous damping of the DET modules [39], but it hampers their ability to return stored energy, and it makes precise force control more challenging. The viscoelastic behavior in our DETs could be greatly reduced by using silicone for their dielectric elastomer [40,41] because silicones typically have much less viscosity than the acrylic elastomer used in this work. Because of their reduced viscosity, silicones have a greater response bandwidth than acrylics [40,41], so they could yield faster stiffness changes than demonstrated here, especially for larger elongations. Silicones typically have a smaller stretch capacity than VHB 4910, but the actuation stretch ($\lambda_r/\lambda_p$) of cone diaphragms (33% in this work) is achievable for silicones because they need less prestretch than VHB 4910 (merely 20% prestretch was used in reference [28]).

Our DET VSA's stiffness change is smaller than that of state-of-the-art VSAs [25], but it could be increased by changing the elastomer used in its DET modules. An elastomer with greater relative permittivity could generate a larger stiffness reduction as seen from Equation (9). This possibility and others have been extensively researched [42].

The high voltages that DET modules require can be difficult to supply in mobile applications. However, small (<2 cm$^3$) DC-DC power converters are commercially available [43], and researchers are developing power supplies optimized for DETs [17,44]. The stiffness variation of a DET is governed by the electric field across it, so DETs with thinner dielectric elastomers can operate at lower voltages. The acrylic material used in this work is also available in half of the thickness that we used (VHB 4905 from 3M), so these devices could be easily redesigned to operate at half of the voltage they use now, though twice as many of them would be required for a given stiffness. Other work has demonstrated DET operation at as low as 300 V with a thinner elastomer [45]. These advancements are paving the way to practical and higher-performance DET devices.

## 5. Conclusions

The DET VSA achieved VSA functionality by using a hybrid actuation architecture: an electric motor modulates the actuator's equilibrium position, and the DET variable-stiffness mechanism modulates the actuator's stiffness. Such decoupled modulation is common in state-of-the-art VSAs but has not been demonstrated with DETs before. This resulting actuator has merely a single component

motion in its variable-stiffness mechanism, giving it a mechanical complexity similar to that of series elastic actuators. This simplicity could make it practical to obtain the benefits of variable-stiffness actuation without the weight, volume, and cost that normally accompany them, once weaknesses of DET technology are addressed.

**Author Contributions:** conceptualization, D.P.A., E.B., W.V., R.D.G.; prototyping and construction, D.P.A. and S.F.; analysis, D.P.A. and E.B.; funding acquisition, W.V. and R.D.G.; investigation, D.P.A. and E.B.; methodology, D.P.A. and E.B.; resources, W.V. and R.D.G.; software, D.P.A. and E.B.; supervision, W.V. and R.D.G.; visualization, D.P.A.; writing—original draft, D.P.A.; writing—review & editing, D.P.A., E.B., S.F., W.V. and R.D.G.

**Funding:** This work was partially supported by the National Science Foundation under award number 1830360. R. D. Gregg holds a Career Award at the Scientific Interface from the Burroughs Wellcome Fund. W. Voit acknowledges support from the DARPA Young Faculty Award, the DARPA Director's Fellowship (No. D13AP00049), and the Center for Engineering Innovation.

**Acknowledgments:** The authors thank (1) Faith Allen and Hannah "Zel" Barber for assistance in assembling DET modules; (2) Wayne Henricks for assistance measuring leakage currents; (3) Forever Faithful Photography for the photos, and (4) Cynthia A. Brewer for providing the plot color schemes through http://www.colorbrewer2.org.

**Conflicts of Interest:** The authors declare no conflict of interest. The funders had no role in the design of the study; in the collection, analyses, or interpretation of data; in the writing of the manuscript, or in the decision to publish the results.

## Abbreviations

The following acronyms are used in this manuscript:

DET   Dielectric elastomer transducer
VHB   Very high bond. A line of adhesive tapes produced by 3M
UTM   Universal testing machine
VSA   Variable-stiffness actuator

## Appendix A. Materials and Methods

### Appendix A.1. DET Materials

Because the purpose of this project was to demonstrate an application of DET technology rather than improve it, we used DET materials that simplified manufacturing rather than those that might provide maximum performance. The dielectric elastomer was VHB 4910, an acrylic elastomer that is sold commercially by 3M as a double-sided adhesive tape. VHB 4910 is 1 mm thick, but we stretch it by hand to 400% biaxial stretch during construction, which thins it to about 63 µm making it more responsive to lower voltages. The electrode material was graphite nanopowder (US Research Nanomaterials # US1058), and it was applied by hand with a sponge applicator. Strips of conductive fabric (Less EMF Stretch Conductive Fabric, # A321) were used to make electrical contact with the graphite electrodes. The module frames and center disks were 3D printed from Stratasys ABS-M30 on a Stratasys Fortus 400mc 3D printer. The reinforcement material was 76.2 µm (3 mil) thick polyimide adhesive tape, cut to shape with a laser cutter, and applied to the frames and center disks by hand. It extended 2.54 mm (0.1 in.) from the frame into the electrode region. The materials used to make a single module cost less than $5.

### Appendix A.2. Testbed

We used a custom-built universal testing machine (UTM) (depicted in Figure A1) to demonstrate and quantify our VSA's capabilities. Its frame is made from aluminum T-slot tubing. This test bed has a linear actuator (PC32LX, Thomson Industries, Radford, VA, USA powered by a BF34-200, Magmotor Technologies, Worcester, MA, USA) which we refer to as the load actuator, that is connected to the output point of our VSA through a load cell (SM2000, Interface, Scottsdale, AZ, USA). During experiments, the load actuator displaced the VSA's output point, and the load cell measured the force output of the VSA.

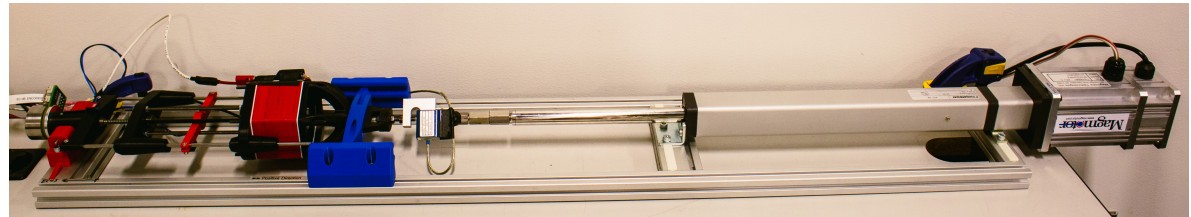

**Figure A1.** The custom universal testing machine (UTM) used for testing is depicted here with the DET VSA connected and the 200 N load cell installed.

The testbed's microcomputer (myRIO 1900, National Instruments, Austin, TX, USA) controls the UTM and the DET VSA, and records sensor data. The myRIO communicates with two motor controllers (EPOS2 70/10, Maxon Precision Motors, Taunton, MA, USA) with the CAN protocol and these motor controllers control the UTM motor and the VSA motor. Using analog signals, the myRIO controls the high-voltage power supply (DCH3320P1, Dean Technology, Dallas, TX, USA) that powers the VSA's variable-stiffness mechanism. Since the UTM was rigidly connected to the VSA's output point, the position of this point was measured with the encoder on the UTM motor. The load cell signal passed through a signal conditioner (DMA2, Interface, Scottsdale, AZ, USA) and was then read by the myRIO through an analog input. The myRIO recorded the analog signal for the output voltage and current of the high-voltage power supply. During the tensile and VSA signature tests, the sample frequency for all measurements was 50 Hz. During the relaxation and stiffness changing speed tests, the analog signals were sampled at 2 kHz. During post processing, all signal data was filtered with a low-pass filter with a 40 Hz cutoff. All testing was performed at ambient temperature and humidity.

*Appendix A.3. Test Procedures*

The tensile tests resembled cyclic tensile tests, which are used to determine stress-strain behavior of material samples. In these tests, the UTM cyclically compressed the DET VSA 25 mm, stretched it 25 mm, and returned it to its equilibrium length three times, while the DET VSA's motor fixed the VSA's equilibrium position at zero. The motion was performed at constant speeds of [1, 10, 100] mm/s and paused at the extreme points for [20, 2, 0.2] s for the speeds, respectively.

Our DET VSA's ability to independently modulate its stiffness and equilibrium position was tested with a modified tensile test. In this test, the motion cycles traversed 15 mm of compression and tension instead of 25 mm to keep the overall motion within the VSA's travel limits. The cycles were conducted at 10 mm/s with 1.2 s of dwell time at each extreme position. The first cycle was performed with the DET modules discharged, then the motion paused while the modules were charged to 5.0 kV, and then a second cycle was performed. The modules were discharged and the VSA's motor shifted the VSA's equilibrium 30 mm from equilibrium position 1 to equilibrium position 2 while the UTM kept the variable-stiffness mechanism unstretched. Finally, another pair of motion cycles like the first were performed.

We used the relaxation tests, so called because they resemble relaxation studies for viscoelastic materials [46], to determine the amount of energy required to soften the variable-stiffness mechanism. In these tests, the VSA's motor was commanded to maintain the equilibrium position at zero, and the UTM held the output at a fixed position. Then, the high-voltage power supply charged the DET modules causing the variable-stiffness mechanism to soften, while the output current and voltage of the high-voltage power supply were recorded. The power supply maintained the charge for 10 s, which allowed the DET modules to charge fully. The input power for the variable-stiffness mechanism was calculated as the product of the high-voltage power supply's output voltage and current. The energy required to soften the variable-stiffness mechanism was the integral of its input power from the start of the experiment until the input power dropped to its steady-state level. Trials were performed with combinations of [5, 15, 25] mm of stretch, and [2, 4, 5] kV of voltage. To keep the output current of the high-voltage power supply to less than 5 mA and not exceed its capabilities, the voltage on the

DET modules was ramped for a period of 0.25 s for the [2 and 4] kV trials and 0.5 s for the 5.0 kV trials before being held steady.

Because the high-voltage power supply's current capability limited the charging speed during the relaxation tests, an additional set of relaxation tests were performed to demonstrate how fast our DET VSA could soften. In these tests, only three DET modules were installed in the variable-stiffness mechanism, without the insulating modules that would dilute the stiffness change. The 2 kN load cell was replaced with a 200 N load cell (SM200, Interface, Scottsdale, AZ, USA) to increase measurement sensitivity. The variable-stiffness mechanism was charged to 5.0 kV with displacements of [5, 15, 25] mm.

The power to maintain a stiffness reduction was investigated by measurements of the leakage current through a set of five DET modules connected in parallel. The leakage current through the five DET modules was measured by connecting them to a high-voltage power supply (EQ-30P1, Matsusada, 745 Aojicho Kusatsu, Shiga 525-0041, Japan) with a multimeter (AN8008, ANENG, Shenzhen, China) in ammeter mode in series between the DET modules and ground because the current measurements from the DCH3320P1 high-voltage power supply's current monitor were not precise enough. The voltage was ramped manually to 5 kV, and then current readings from the multimeter were recorded every 30 s for 6 min. The voltage was then ramped to 0 kV, and after a waiting period of a few minutes, the process was repeated with [4 and 2] kV.

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
