# Peer review of "Mechanical Simplification of Variable-Stiffness Actuators Using Dielectric Elastomer Transducers"

_actuators, doi:10.3390/act8020044_

Round 1

Reviewer 1 Report

The authors designed a variable stiffness actuator by using the dielectric elastomers. Such actuators have the potential application in soft robots. The comments are as follows:

1. The reviewer does not totally agree with the relation between force and true stress as presented in Eq. (4). In fact, the radial stretch in the deformed dielectric elastomer films is inhomogeneous. At the deformed state, the radial stretch at the inner edge is really different with that at the outer edge. That is, the thickness of the deformed dielectric elastomer film is inhomogeneous, instead of homogeneous. 

2. In Eq. (6), why the term of electrical contribution is not included? 

3. It seems that Eq. (7) is not correct. Please check!

4. What expressed in Table 2 is not clear. 

5. The stiffness of the actuator should be clearly defined.

If the raised comments have been well responded, the manuscript can be accepted for publication.

Author Response

a

Reviewer 2 Report

The reviewer appreciates thorough descriptions of the procedures, investigation of necessary aspects, and detailed explanations as well as well-presented results. Most parts are carefully discussed and there are not much to add or subtract. One drawback could be its less novelty using cone diaphragms with an electric motor but this seems tenable as the focus is developing variable stiffness DET actuators.

Author Response

Uploaded as PDF file.

Reviewer 3 Report

The paper presents a quite well-designed prototype that integrates a conical-shaped dielectric elastomer transducer that is employed as an electrically variable stiffness element. 

The paper is also sufficiently well written, but if it would need a review to simplify sentences constructions and writings. 

The main issue with this paper is that the same concept/architecture implemented with the same materials and very similar methods have been conceived by Prof. Cutkosky in Stanford. 

The only claimed novelty is that the proposed seems to be related to the capability of the system to modulate simultaneously equilibrium position and stiffness. But this is an obvious feature of any variable stiffness element that is connected in series with an actuator. 

I can not see any other novel contribution in these paper with respect to the paper listed below, thus I can not recommend this paper for the publication. One option for the authors might be to repeat the experiments using at least different materials as dielectrics and resubmit their work. 

PAPERS:

Dastoor, S.; Cutkosky,M.R. Design of Dielectric Electroactive Polymers for a Compact and Scalable Variable Stiffness Device. IEEE International Conference on Robotics and Automation; IEEE: Saint Paul, 2012; pp. 3745–3750.

Dastoor, Sanjay K., and Mark R. Cutkosky. "Electrically variable suspension." U.S. Patent 9,765,837, issued September 19, 2017.

Orita, A.; Cutkosky, M.R. Scalable Electroactive Polymer for Variable Stiffness Suspensions. IEEE/ASME, Transactions on Mechatronics 2016, 21, 2836–2846.

Author Response

Uploaded as PDF file

Round 2

Reviewer 3 Report

The authors have now clarified what is the novel content of the manuscript. I still believe that the innovation content not major. However, the provided results represent a complete and good quality example of application of a previously conceive concept.  

The quality of writing and organisation of the contents are also acceptable, but please consider a further a revision of the writing style in order to improve/simplify the construction of sentences.